materials science/inorganic chemistry/
nanotechnology

WO₃, nanoparticles, biomimetic synthesis,
aqueous solution process

**Author for correspondence:**
Hiroaki Uchiyama
e-mail: h_uchi@kansai-u.ac.jp

This article has been edited by the Royal Society of Chemistry, including the commissioning, peer review process and editorial aspects up to the point of acceptance.

# Biomimetic synthesis of nanostructured WO₃·H₂O particles and subsequent thermal conversion to WO₃

Hiroaki Uchiyama, Shouta Mizuguchi and Shiho Hirano

Department of Chemistry and Materials Engineering, Kansai University, 3-3-35 Yamate-cho, Suita 564-8680, Japan

HU, 0000-0001-9337-6418

Nanostructured tungsten oxide (WO₃) particles were prepared in aqueous solution by mimicking biomineralization. Precursor WO₃·H₂O particles were generated by ageing a 60°C (NH₄)₁₀W₁₂O₄₁·5H₂O solution containing gelatin. This was followed by heating to 600°C in air for thermal conversion to WO₃. The addition of gelatin led to the formation of layered structures consisting of WO₃·H₂O platy particles, which contained segmented, block-like nanoscale units. The macroscopic layered structure was preserved after thermal conversion to WO₃, while the morphology of the block-like units changed to orthogonally crossed nanorods.

## 1. Introduction

Tungsten oxide (WO₃) has been used in various devices such as photoelectrodes [1–3], electrochromic materials [1,4,5] and gas sensors [6–8]. Device performance is affected by crystallite size and shape and the morphology of the secondary particles. Thus, nanostructural control of WO₃ crystals is important for practical applications.

Recently, various nanostructured inorganic materials have been prepared by novel solution techniques that mimicked biomineralization. Natural biominerals such as nacres, sea urchin spines and eggshells consist of oriented inorganic units less than 50 nm in size, and nanoscale spaces containing biological polymers like chitin, chitosan and gelatin [9–17]. Such highly ordered nanostructures are formed by self-assembly and self-organization via interactions between the inorganic crystals and the biological polymers. The polymers contain many polar amino acids [18] that adsorb on the surfaces of the inorganic crystals, creating hierarchical structures consisting of nanoscale crystallites. Biomimetic structures have been widely made from nanoscale inorganic units and biological polymers [11–13,18–25]. Many

works about the biomimetic synthesis of functional metal oxide materials mainly focused on the similarity in the resultant nanostructure between the products and the real biominerals, and the resultant device performance. On the other hand, we have focused on 'biomimetic synthetic route' and attempted to construct novel aqueous techniques for making nanostructured materials. We think that the key factors of biomineralization are (i) the interaction between inorganic crystals and biological polymers, and (ii) the multistep synthetic procedure via metastable phases as the precursor materials, and have suggested new approaches containing one or both factors for making hierarchical structures consisting of oriented nanocrystallites like biominerals. Cocoon-like $CeCO_3OH$ particles consisting of nanoscale crystallites were prepared from aqueous solutions and gels containing $CeCl_3$ and biological polymers such as gelatin and agar, by the addition of $(NH_4)_2CO_3$ solutions. They were then thermally converted to $CeO_2$ particles with the same morphologies [26]. Spherical SnO particles consisting of radially branched platy units were produced by ageing $Sn_6O_4(OH)_4$ in aqueous solutions containing gelatin at 60°C [27]. Hence, biomimetic aqueous routes are promising ways to fabricate nanostructured inorganic materials.

In this work, nanostructured $WO_3$ particles were prepared by a biomimetic aqueous solution process with gelatin. Tungsten oxides can exist in aqueous solutions as monomeric tungstate ions ($WO_4^{2-}$) or para-tungstate ions ($HW_6O_{21}^{5-}$, $H_2W_{12}O_{42}^{10-}$, etc.) [28–32]. The ions can be precipitated as hydrous tungsten oxides ($WO_3 \cdot xH_2O$) [33,34], tungstate ($H_2WO_4$, $H_4WO_5$, etc.) [35,36] crystals or $WO_3$. The crystal phase and morphology of the precipitates are affected by the pH and by the concentrations of the precursors. Here, nanostructured $WO_3 \cdot H_2O$ particles were prepared as $WO_3$ precursors from $(NH_4)_{10}W_{12}O_{41} \cdot 5H_2O$ aqueous solutions that contained gelatin; the $WO_3$ particles were subsequently obtained by heating the precursors. As mentioned above, we have previously tried to prepare nanostructured $CeO_2$ materials on the basis of a similar strategy [26]. In that work, nanostructured $CeCO_3OH$ particles were obtained with biological polymers and then thermally converted to $CeO_2$, while the crystallographic orientation of inorganic units like biominerals was not observed in the $CeO_2$ products [26]. On the other hand, $WO_3 \cdot H_2O$ were reported to topotactically transform to monoclinic $WO_3$ crystals [33], which would allow us to keep the crystallographic orientation of inorganic units after the thermal conversion to metal oxide. We varied the pH and the gelatin concentration to investigate the effects on size, shape and crystal phase of the $WO_3$ precursors and $WO_3$ particles.

# 2. Material and methods

Aqueous HCl solutions with pH 0.6–1.0 were prepared by diluting 36.0 mass % hydrochloric acid (Wako Pure Chemical Industries, Osaka, Japan) with purified water. Then, 0.10 g of $(NH_4)_{10}W_{12}O_{41} \cdot 5H_2O$ (Wako Pure Chemical Industries, Osaka, Japan) was dissolved in 20 cm³ of the HCl solutions by stirring at 80°C for 3 min. When 0–0.040 g of gelatin (Wako Pure Chemical Industries, Osaka, Japan) was added, the solutions immediately became cloudy. After stirring at 80°C for 3 h, the cloudy suspensions became transparent and were then used as the precursor solutions ($[(NH_4)_{10}W_{12}O_{41} \cdot 5H_2O] = 1.7$ mM, [gelatin] ($C_{ge}$) = 0–2.0 g l⁻¹). These solutions were aged at 60°C for 1–7 days, resulting in yellowish precipitates that were washed with purified water and dried at 60°C for 24 h. The precipitates were $WO_3$ precursors that were heated at a rate of 5°C min⁻¹ to 600°C, which was maintained at the heating temperature (600°C) for 24 h in the air for conversion to $WO_3$.

The crystalline phases of the $WO_3$ precursors and the heat-treated $WO_3$ products were identified by X-ray diffraction (XRD) in an ordinary $2\theta/\theta$ mode, with a CuK$\alpha$ X-ray diffractometer (Model Rint 2550V, Rigaku, Tokyo, Japan) operated at 40 kV and 300 mA. The microstructures of the precursors and the heat-treated samples were imaged with a field-emission scanning electron microscope (FE-SEM) (Model JSM-6500F, JEOL, Tokyo, Japan) and a field-emission transmission electron microscope (FE-TEM) (JEM-2000EX, JEOL, Tokyo, Japan). Thermogravimetric and differential thermal analysis (TG–DTA) curves were obtained for the $WO_3$ precursors at a heating rate of 10°C min⁻¹ in flowing air with a thermal analyser (Model ThermoPlus 2, Rigaku, Tokyo, Japan).

# 3. Results and discussion

## 3.1. Preparation of $WO_3$ precursors

At first, we performed preliminary experiments to know the reaction time at which the increase in the sample yield stopped. Aqueous solutions of 1.7 mM $[(NH_4)_{10}W_{12}O_{41} \cdot 5H_2O]$ and 0–2.0 g l⁻¹ gelatin

**Table 1.** Ageing times and precursor yields.

| pH of solvents | [gelatin] (g l$^{-1}$) | ageing time (day) | yield (%) |
|---|---|---|---|
| 0.6 | 0 | 1.0 | 63.3 |
| 0.6 | 0.1 | 1.0 | 30.2 |
| 0.6 | 0.2 | 1.1 | 24.0 |
| 0.6 | 0.5 | 3.6 | 28.0 |
| 0.6 | 1.0 | 5.8 | 31.3 |
| 0.6 | 1.5 | 5.0 | 25.7 |
| 0.6 | 2.0 | 7.0 | 30.0 |
| 0.8 | 0 | 1.0 | 47.2 |
| 0.8 | 0.1 | 1.1 | 33.2 |
| 0.8 | 0.2 | 2.0 | 25.6 |
| 0.8 | 0.5 | 5.0 | 18.6 |
| 0.8 | 1.0 | 5.0 | 39.3 |
| 0.8 | 1.5 | 5.7 | 26.1 |
| 0.8 | 2.0 | 5.0 | 21.2 |
| 1.0 | 0 | 1.0 | 55.2 |
| 1.0 | 0.1 | 4.0 | 25.4 |
| 1.0 | 0.2 | 4.0 | 23.4 |
| 1.0 | 0.5 | 5.0 | 39.5 |
| 1.0 | 1.0 | 7.0 | 22.0 |
| 1.0 | 1.5 | 7.0 | 22.3 |
| 1.0 | 2.0 | 7.0 | 18.3 |

($C_{ge}$) with HCl at pH 0.6–1.0 were aged at 60°C for 1–7 days. Yellowish WO$_3$ precursors were precipitated by ageing irrespective of $C_{ge}$ and pH. The ageing times at which the increase in the sample yield stopped, and precursor yields are listed in table 1.

The precipitation of the WO$_3$ precursors was slower and the yield decreased with increasing pH, which indicated that nucleation was suppressed by the decreased acidity. Tungsten oxides precipitate as hydrous tungstic acid (H$_2$WO$_4$ · $n$H$_2$O) [35,36] and tungsten trioxide (WO$_3$ · $n$H$_2$O) [33,34] under strongly acidic conditions, and their solubility increases with pH [28–32]. In the present case, the higher solubility under more weakly acidic conditions caused a slower nucleation rate and thus a lower yield of WO$_3$ precursors. Moreover, the addition of gelatin also inhibited the deposition of WO$_3$ precursors because its amino groups might have coordinated with tungstate ions, leading to suppressed nucleation. On the basis of these results, we employed the ageing time described in table 1 for sample preparation. Figure 1 shows XRD patterns of the WO$_3$ precursors. The diffraction peaks attributed to WO$_3$ · H$_2$O were observed irrespective of the pH and the gelatin concentration in the precursor solutions. The peak intensities of the (020) plane of the precursors prepared with gelatin were higher than those in the powder diffraction file (WO$_3$ · H$_2$O: PDF#43-0679). The precipitation of WO$_3$ · H$_2$O under an acidic condition seems to be as follows:

$$WO_4{}^{2-} + 2H^+ \rightarrow WO_3 \cdot H_2O. \tag{3.1}$$

This reaction consumes H$^+$ ions, resulting in the increase in the pH value. On the other hand, in the present case, the pH value is almost unchanged after the precipitation. Here, the [(NH$_4$)$_{10}$W$_{12}$O$_{41}$ · 5H$_2$O] was very low (1.7 mM), and thus the pH change was deduced to be small during the reaction.

Figure 2 shows SEM images of WO$_3$ precursors prepared from (NH$_4$)$_{10}$W$_{12}$O$_{41}$ · 5H$_2$O solutions with HCl at pH 0.6–1.0 without gelatin ($C_{ge}$ = 0 g l$^{-1}$). Random aggregates of platy particles 1–2 μm in width were obtained irrespective of the pH. The morphology of the WO$_3$ precursors changed with the addition of gelatin, and its effect varied with the pH. SEM microstructure images of WO$_3$ precursors prepared with gelatin ($C_{ge}$ = 0–2.0 g l$^{-1}$) are shown in figures 3 and 4, for (NH$_4$)$_{10}$W$_{12}$O$_{41}$ · 5H$_2$O solutions with

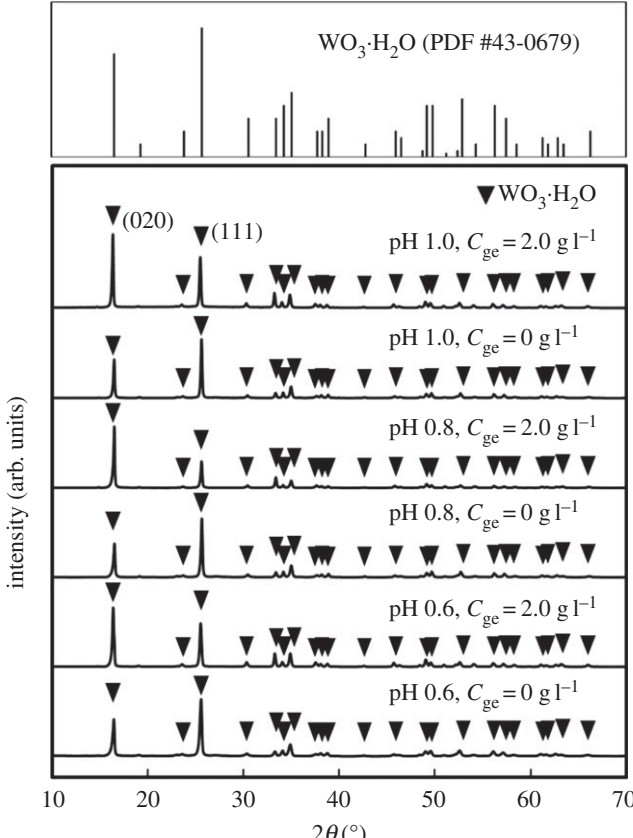

**Figure 1.** XRD patterns of $WO_3$ precursors prepared from $(NH_4)_{10}W_{12}O_{41}$ solutions with $C_{ge} = 0-2.0$ g l$^{-1}$ and HCl at pH $0.6-1.0$ (the ageing time was as shown in table 1).

HCl at pH 0.6 and 1.0, respectively. Under relatively strong acidic conditions (pH 0.6), layered structures with widths of 10 µm and thicknesses of 5–10 µm appeared when $C_{ge} = 0.2$ g l$^{-1}$ (figure 3$a$,$b$); the structures consisted of stacked platy units. For $C_{ge} > 0.5$ g l$^{-1}$, the plate-like microstructure collapsed, resulting in unshaped aggregates with inhomogeneous sizes and shapes (figure 3$c$,$d$). Under weakly acidic conditions (pH 1.0), layered structures were obtained by the addition of gelatin (figure 4$a$), as well as more acidic conditions (pH 0.6) (figure 3$a$,$b$). In weakly acidic conditions, the addition of large amounts of gelatin did not result in the collapse of the platy structure. The increase to $C_{ge} = 1.5$ g l$^{-1}$ induced the formation of large layered plates with widths of 20–50 µm and thicknesses of 10–20 µm (figure 4$b$,$c$), with relatively homogeneous sizes and shapes. Moreover, segmented block-like units of 0.50–1 µm in size were observed on the side faces of the plates (figure 4$d$), which suggested that large amounts of gelatin caused branching of the platy particles.

We investigated the effect of the ageing time on the morphology and crystal phase of $WO_3$ precursors. $WO_3$ precursors were prepared by ageing for 1–7 days from $(NH_4)_{10}W_{12}O_{41}$ solutions with $C_{ge} = 2.0$ g l$^{-1}$ and HCl at pH 1.0. No precipitation was observed for 1–3 days, while precipitates appeared after 4 days. Figure 5 shows the XRD patterns of $WO_3$ precursors prepared by ageing for 4–7 days. The diffraction peaks attributed to $WO_3 \cdot H_2O$ were observed irrespective of the ageing times. Figure 6 shows the SEM images of the $WO_3$ precursors. The precipitate obtained on 4 days was the mixture of layered plates and spherical particles (figure 6$a$). The spherical particles disappeared with increasing ageing times, and only layered plates were observed after 6 days (figure 6$b$). The spherical particles found in the precipitates 4–5 days were thought to be the composites of tungstate ions and gelatin. As described in the experimental section, in this work, the $(NH_4)_{10}W_{12}O_{41}$ aqueous solutions immediately became cloudy on addition of gelatin, which might be attributed to the formation of the composites of tungstate ions and gelatin. The cloudy suspension became transparent again by stirring at 80°C and then was used as the precursor solutions. In the case of the 4–5 days ageing, the precipitation of $WO_3 \cdot H_2O$ did not complete, and thus unreacted tungstate ions remained in the solutions. The tungstate ions might precipitate as the gelatin composite during cooling, forming the spherical particles.

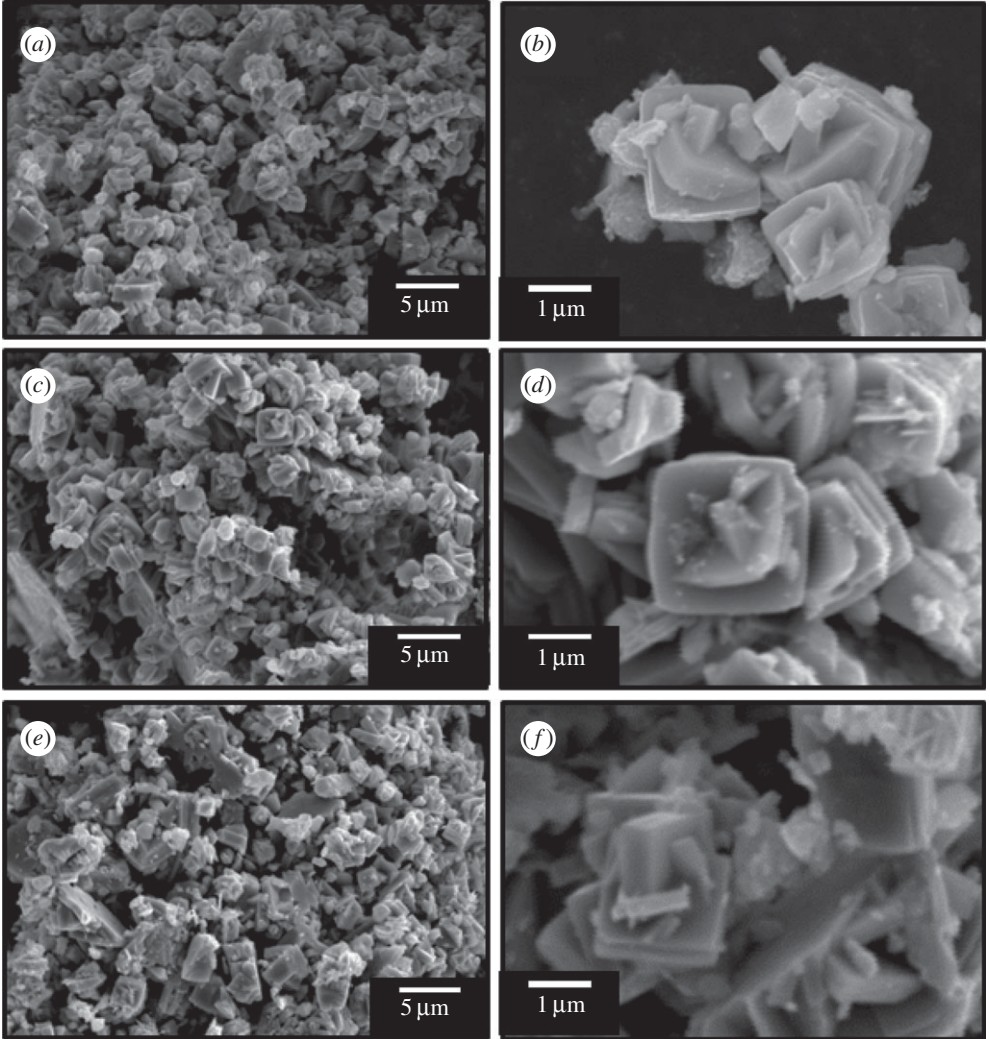

**Figure 2.** SEM images of WO$_3$ precursors prepared from $(NH_4)_{10}W_{12}O_{41}$ solutions with $C_{ge} = 0$ g l$^{-1}$ and HCl at pH 0.6 (*a,b*), pH 0.8 (*c,d*) and pH 1.0 (*e,f*) (the ageing time was as shown in table 1).

In the present work, the addition of gelatin led to the formation of WO$_3 \cdot$ H$_2$O layered structures (figures 3 and 4). The WO$_3$ precursors prepared with gelatin exhibited an intense diffraction peak from the (020) plane (figure 1), which indicated that the flat face of the layered structures was the (010) plane of WO$_3 \cdot$ H$_2$O crystals. This agreed well with previous reports that the synthesis of WO$_3 \cdot$ H$_2$O plates exposed the (010) plane as the flat face [7,33,37]. The morphological change of the WO$_3$ precursors from random aggregates of platy particles (figure 2) to layered structures (figures 3 and 4) was attributed to the adsorption of gelatin on the (010) planes of the WO$_3 \cdot$ H$_2$O crystals. Without gelatin, heterogeneous nucleation rapidly occurred on the surface of WO$_3 \cdot$ H$_2$O platy particles and crystallites grew in various directions, resulting in random aggregates (figure 2). Alternatively, gelatin might have adsorbed on the (010) plane of the WO$_3 \cdot$ H$_2$O crystals and suppressed nucleation and crystal growth on the surface of the plates. The mild nucleation and growth rates could have caused the slow growth of new platy crystallites along the flat face of the WO$_3 \cdot$ H$_2$O plates, resulting in layered structures (figure 3*a,b* and figure 4).

Moreover, weakly acidic conditions (pH 1.0) resulted in large layered plates with more homogeneous sizes and shapes (figure 4*b*). This could have been caused by the slower nucleation rate because of the higher solubility of the tungsten compounds. In such a case, the branching growth of platy particles occurred with increasing gelatin concentration (figure 4*d*). The larger amounts of gelatin could adsorb on the side face of WO$_3 \cdot$ H$_2$O plates as well as the flat face, which inhibited the growth in the lateral direction. This resulted in the branching of platy particles and the subsequent formation of the segmented block-like units (figure 4*d*).

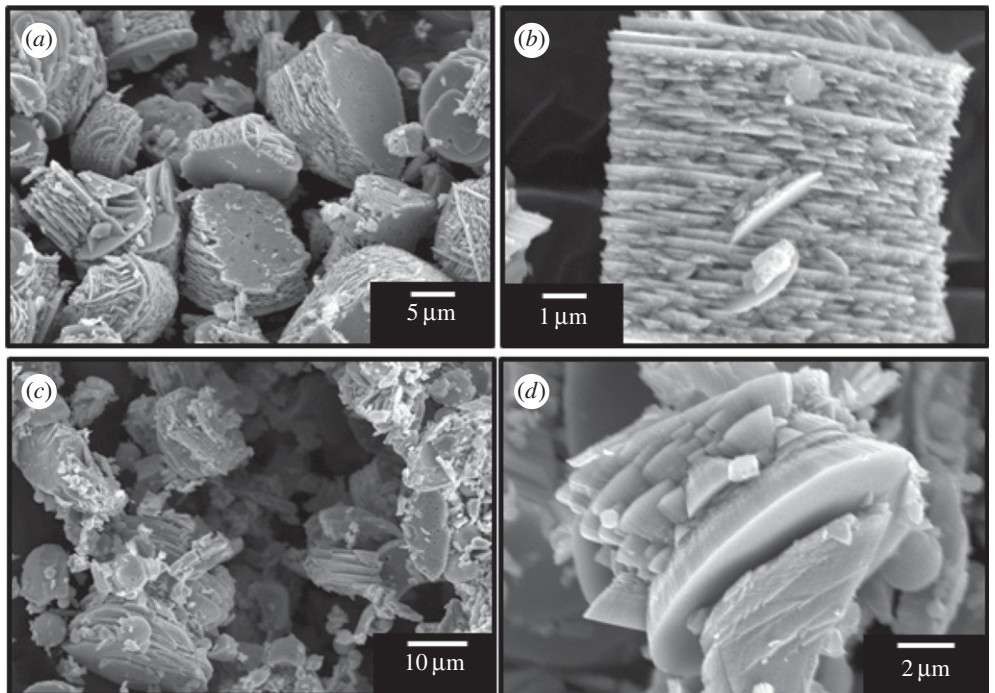

**Figure 3.** SEM images of $WO_3$ precursors prepared from $(NH_4)_{10}W_{12}O_{41}$ solutions with $C_{ge} = 0.2$ g l$^{-1}$ (*a,b*) and 2.0 g l$^{-1}$ (*c,d*) and HCl at pH 0.6 (the ageing time was as shown in table 1).

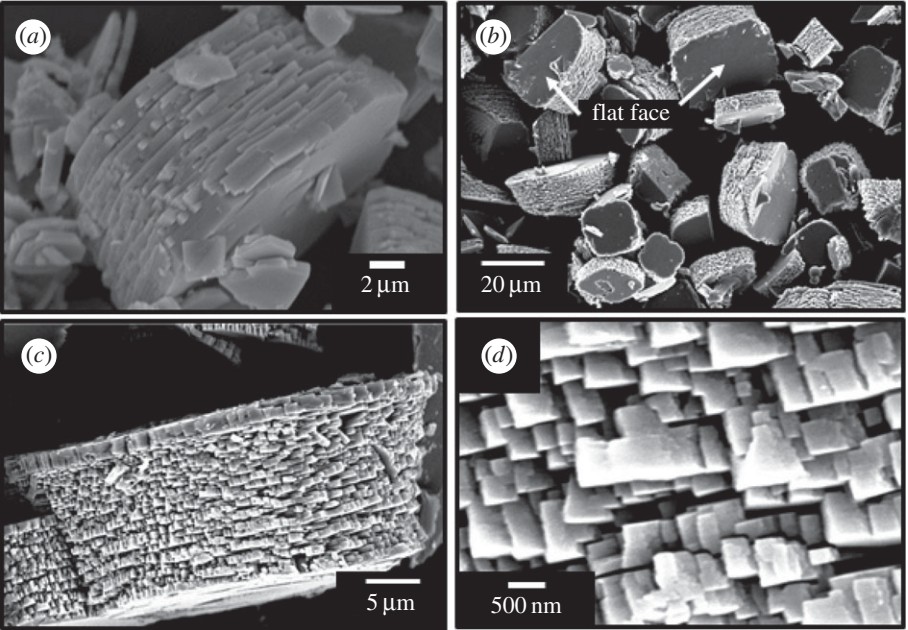

**Figure 4.** SEM images of $WO_3$ precursors prepared from $(NH_4)_{10}W_{12}O_{41}$ solutions with $C_{ge} = 0.2$ g l$^{-1}$ (*a*) and 1.5 g l$^{-1}$ (*b–d*) and HCl at pH 1.0 (the ageing time was as shown in table 1).

## 3.2. Thermal conversion to $WO_3$ particles

The $WO_3$ precursors ($WO_3 \cdot H_2O$) obtained with gelatin were heated for thermal conversion to $WO_3$. Figure 7 shows TG–DTA curves for the $WO_3$ precursors ($C_{ge} = 2.0$ g l$^{-1}$, pH 1.0). The first weight loss with an endothermic peak at 220°C indicated dehydration and was close to the theoretical weight loss (7.2 wt%) of the reaction

$$WO_3 \cdot H_2O \rightarrow WO_3 + H_2O. \quad (3.2)$$

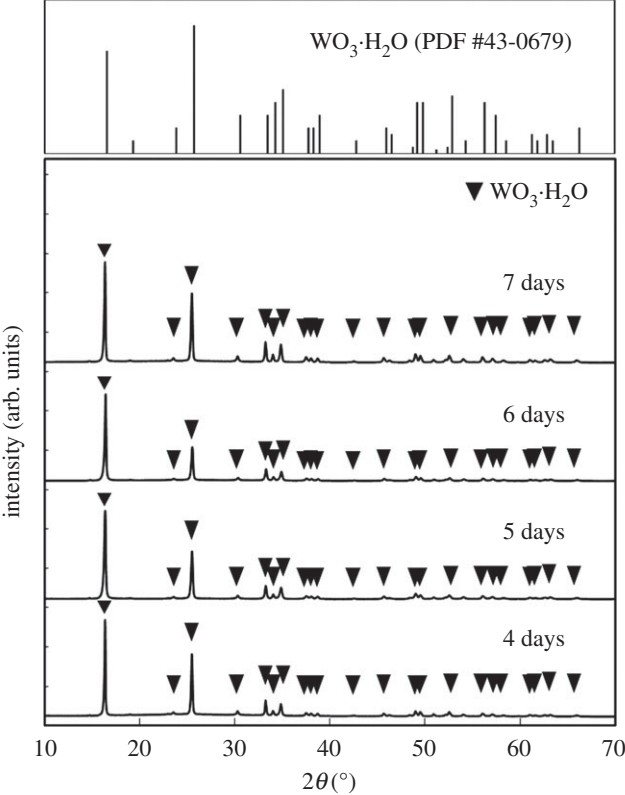

**Figure 5.** XRD patterns of $WO_3$ precursors prepared by ageing for 4–7 days from $(NH_4)_{10}W_{12}O_{41}$ solutions with $C_{ge} = 2.0$ g l$^{-1}$ and HCl at pH 1.0.

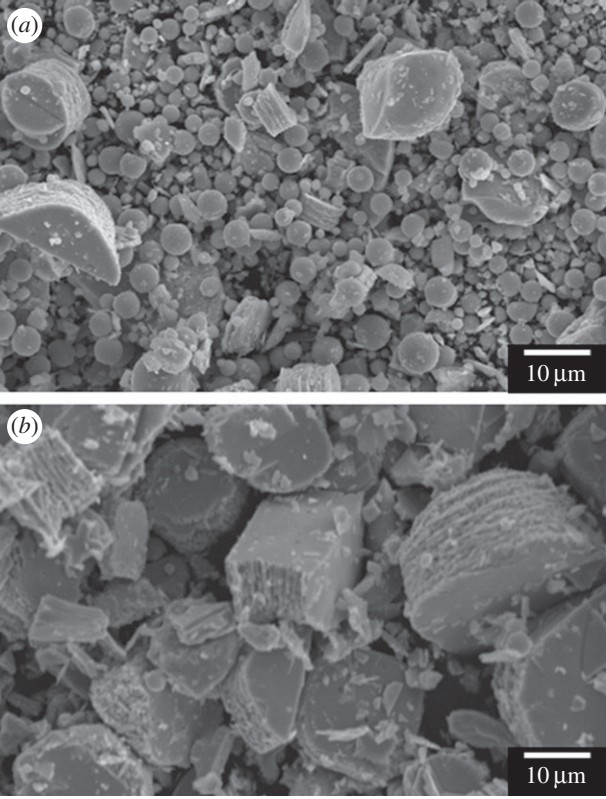

**Figure 6.** SEM images of $WO_3$ precursors prepared by ageing for 4 (a) and 6 (b) days from $(NH_4)_{10}W_{12}O_{41}$ solutions with $C_{ge} = 2.0$ g l$^{-1}$ and HCl at pH 1.0.

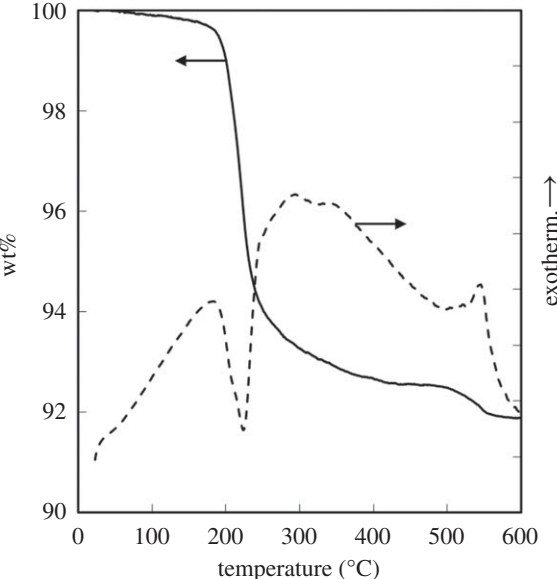

**Figure 7.** TG–DTA curves for $WO_3$ precursors prepared from $(NH_4)_{10}W_{12}O_{41}$ solutions with $C_{ge} = 2.0$ g l$^{-1}$ and HCl at pH 1.0 (the ageing time was as shown in table 1).

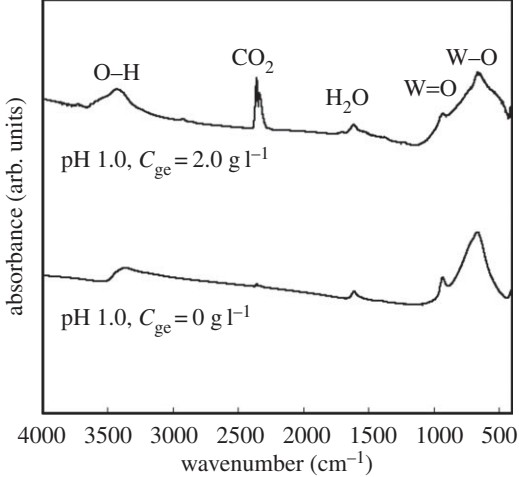

**Figure 8.** FT-IR spectra for $WO_3$ precursors prepared from $(NH_4)_{10}W_{12}O_{41}$ solutions with $C_{ge} = 0$ and $2.0$ g l$^{-1}$ and HCl at pH 1.0 (the ageing time was as shown in table 1).

In addition, a slight weight loss of 1 wt% was detected at 550°C, which could be attributed to the combustion of residual gelatin. We also investigated the presence of residual gelatin on the $WO_3$ precursors by FT-IR analysis. Figure 8 shows FT-IR spectra for the $WO_3$ precursors ($C_{ge} = 0$ and $2.0$ g l$^{-1}$, pH 1.0). No absorption peaks due to gelatin were detected, which also indicates that only a little amount of gelatin remained on the $WO_3 \cdot H_2O$ products.

The large $WO_3 \cdot H_2O$ layered plates ($C_{ge} = 1.5$ g l$^{-1}$, pH 1.0) were converted to $WO_3$ by heat treatment at 600°C for 24 h in air. Figure 9 shows the XRD patterns of the heat-treated $WO_3$. Diffraction peaks attributed to monoclinic $WO_3$ were observed, and the $WO_3 \cdot H_2O$ phase was absent.

Figure 10 shows FE-SEM and FE-TEM images of the heat-treated $WO_3$ products ($C_{ge} = 1.5$ g l$^{-1}$, pH 1.0). As shown in figure 10a, the layered structure of the $WO_3$ precursors remained after thermal conversion to $WO_3$. In addition, orthogonally crossed nanorods 50 nm in width were observed in the $WO_3$ layers (figures 10b,c). Regular diffraction spots were observed in the selected area electron diffraction pattern of the $WO_3$ (figure 10d). The diffraction spots indicated that the rod-like units were oriented in the same crystallographic direction, and the flat face of heat-treated $WO_3$ layers is (001) plane of monoclinic $WO_3$. $WO_3 \cdot H_2O$ were reported to topotactically transform to monoclinic $WO_3$ crystals [33]. As discussed in the XRD measurement of $WO_3$ precursors, the flat face of the $WO_3 \cdot H_2O$ layered structures was the (010) plane of $WO_3 \cdot H_2O$ crystals. These suggest the topotactic

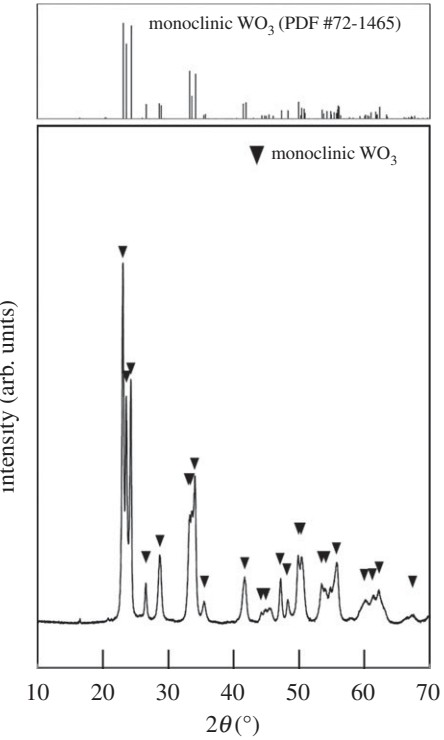

**Figure 9.** XRD patterns of heat-treated $WO_3$ products obtained from $WO_3$ precursors prepared by ageing for 7 days from $(NH_4)_{10}W_{12}O_{41}$ solutions with $C_{ge} = 1.5$ g l$^{-1}$ and HCl at pH 1.0.

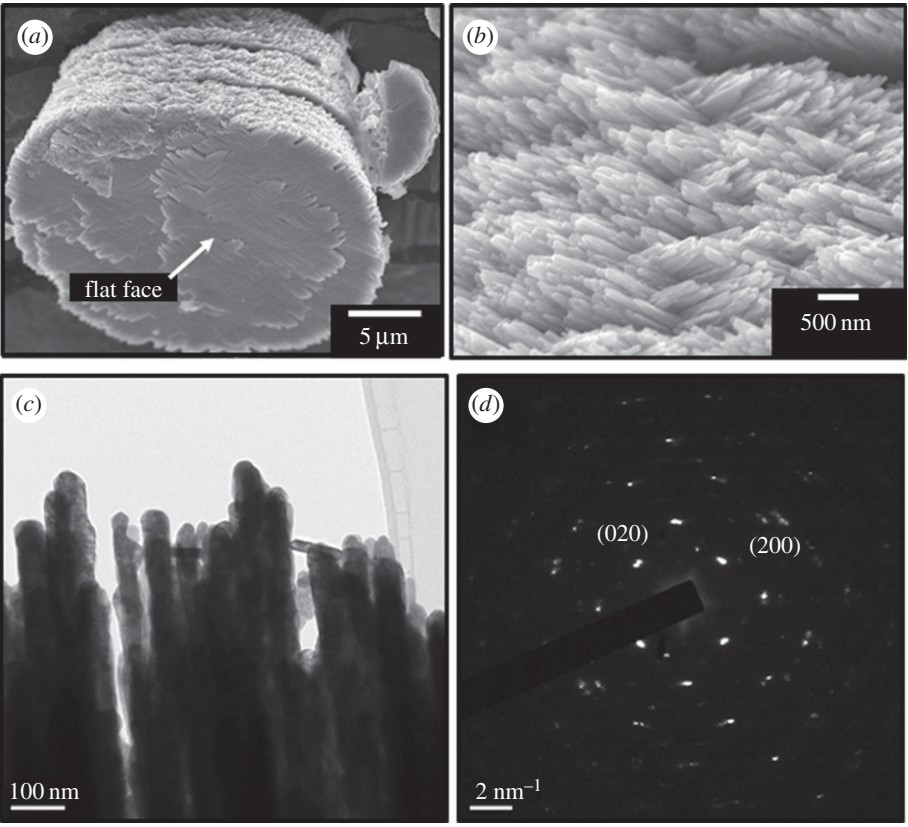

**Figure 10.** FE-SEM (*a,b*) and FE-TEM (*c,d*) images of heat-treated $WO_3$ products from $WO_3$ precursors prepared by ageing for 7 days from $(NH_4)_{10}W_{12}O_{41}$ solutions with $C_{ge} = 1.5$ g l$^{-1}$ and HCl at pH 1.0.

transformation of [010]-oriented $WO_3 \cdot H_2O$ layered plates to [001]-oriented $WO_3$. These results suggested that the $WO_3$ layered structures prepared with gelatin have highly ordered nanostructures consisting of oriented inorganic nanoscale units like biominerals such as nacres, sea urchin spines and eggshells [9–17]. We evaluated the BET surface area of the $WO_3$ products by $N_2$ adsorption method. The surface area of the layered structure obtained by an addition of gelatin ($C_{ge} = 1.5$ g $l^{-1}$, pH 1.0) was $5.02$ $m^2$ $g^{-1}$, which was larger than that of the random aggregates of $C_{ge} = 0$ g $l^{-1}$, pH 1.0 ($2.31$ $m^2$ $g^{-1}$). The hierarchical nanostructures are thought to be suitable for photoelectrode and gas sensor materials.

# 4. Conclusion

$WO_3$ particles with highly ordered nanostructures were prepared via a biomimetic process involving the biological polymer gelatin. $WO_3 \cdot H_2O$ platy particles were $WO_3$ precursors obtained from $(NH_4)_{10}W_{12}O_{41} \cdot 5H_2O$ aqueous solutions, where the addition of gelatin resulted in morphological changes from random aggregates to layered structures. The layered structures consisted of platy particles branching to block-like nanoscale units that were induced by the suppression of nucleation and growth by gelatin adsorption. Nanostructured $WO_3$ particles were obtained from the $WO_3 \cdot H_2O$ layered structures by heat treatment. A morphological change of nanoscale units from segmented blocks to orthogonally crossed nanorods was observed. Overall, these results suggested that an aqueous route mimicking biomineralization was effective for nanostructural control of inorganic materials.

Ethics. *Research ethics:* We do not require any research ethical approval, licence or permission because it is not relevant to our work. *Animal ethics*: We do not require any animal ethical approval, licence or permission because it is not relevant to our work.

Data accessibility. There are no additional data to accompany this manuscript. All relevant data (the table of ageing times and product yields, XRD patterns, SEM and TEM images and TG–DTA curves) are within the main body of the manuscript.

Authors' contributions. H.U. conceived of the study, designed the study and drafted the manuscript; S.M. and S.H. carried out the sample synthesis and characterization. All authors gave final approval for publication.

Competing interests. We have no competing interests.

Funding. No financial support was received for this work.

Acknowledgements. We thank Alan Burns, PhD, from the Edanz Group (http://www.edanzediting.com/ac) for editing a draft of this manuscript, and Prof. Hiroyuki T. Takeshita, Prof. Ryota Kondo, from Kansai University, for their help in measurement of surface area.

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
