## [Reviewer comments · Royal Society Open Science]

Review History

RSOS-182137.R0 (Original submission)

Review form: Reviewer 1

Is the manuscript scientifically sound in its present form?

Yes

Are the interpretations and conclusions justified by the results?

Yes

Is the language acceptable?

Yes

Is it clear how to access all supporting data?

Not Applicable

Do you have any ethical concerns with this paper?

No

Have you any concerns about statistical analyses in this paper?

No

Recommendation?

Accept with minor revision (please list in comments)

Comments to the Author(s)

The present manuscript reports biomimetic synthesis of nanostructured $\text{WO}_3 \cdot \text{H}_2\text{O}$ particles. The synthesis/characterization techniques are performed at a high technical standard, and discussion given in the manuscript are basically sound. In this context, this manuscript is potentially acceptable, according to the publication criteria of Royal Society Open Science. However, I think, further discussions should be added to show clearly the back ground of the present study, and give robustness to the conclusions.

I'd like to add following remarks to be further considered by the authors;

- 1) I wonder if the present manuscript is the first report on the biomimetic synthesis of $\text{WO}_3 \cdot \text{H}_2\text{O}$ (and WO_3) or not. Otherwise, what's a significant advance from the previous biomimetic synthesis of WO_3 -based materials? The advance of the present work should be clearly discussed in the introduction section; significant advance of scientific knowledge is a criteria of acceptance by RSOS.
- 2) The authors describe their previous works on the synthesis of CeO_2 and SnO in the introduction section (the second paragraph in the introduction). The context somehow gives us an impression that the present work had been routinely-done. I suggest to add some comments to describe scientific challenges for the biomimetic synthesis of WO_3 , and how the present system is different from previous ones.
- 3) The crystallization in the present system is highly sensitive to the pH of the solution. pH after the reaction as well as before the reaction should be given to confirm robustness of the discussion; precipitation of crystals is usually accompanied by the change of pH of reaction mixture.
- 4) What is the aging time employed for samples appeared in Figs 1, 2, 3, 4, 5, 6, and 7. The authors describes in the experimental section that they tested aging time 1-7 days. Which aging time was chosen for the discussions in respective figures?
- 5) Reaction kinetics of the crystallization process should be discussed on at least one sample. I'm wondering how the crystallization occurs as a function of reaction time. Is there any induction period for the crystallization? How does the morphology develop with reaction time?
- 6) The weight ratio of gelatin/ WO_3 should be calculated from the TG data of the obtained product. This information gives us an idea on what kind of composite is indeed obtained.
- 7) Fig4: "the flat face of the layered structure" is ambiguous. It should be marked in Fig.4.
- 8) It might be interesting to compare TEM images and diffractions before and after the calcination. Is there any relevance of crystallographic orientation between them?

Review form: Reviewer 2**Is the manuscript scientifically sound in its present form?**

Yes

Are the interpretations and conclusions justified by the results?

Yes

Is the language acceptable?

Yes

Is it clear how to access all supporting data?

Not Applicable

Do you have any ethical concerns with this paper?

No

Have you any concerns about statistical analyses in this paper?

No

Recommendation?

Accept with minor revision (please list in comments)

Comments to the Author(s)

The authors report the new biomimetic synthesis of $\text{WO}_3 \cdot \text{H}_2\text{O}$ particles, which can further be converted to WO_3 particles by heat treatment at higher temperature. The synthesis method and discussion on the mechanisms of the particle formation are technically sound, and the conclusions are supported by experimental data obtained from sufficient number of samples. Therefore the manuscript would be publishable in Royal Society Open Science after addressing minor issues shown below.

The inner structure of the particles should be analyzed. Are the size and shape of the pores, the crystalline size and orientation the same on the surface and inside of the particles? Surface area, pore volume, pore size distribution, true density, particle density, etc. are also importance to be measured for detailed characterization.

In the Introduction section, the authors mentioned that WO_3 has been used in various devices. Thus, readers would be interested in what sort of advantages the newly prepared WO_3 particles have or which application field they can be practically used in.

Compared with the other published works on biomimetic synthesis, the method and findings in the manuscript sound not quite new. Thus, a little more characterization or property measurement of the obtained particles would further demonstrate importance of the work.

Decision letter (RSOS-182137.R0)

29-Mar-2019

Dear Dr Uchiyama:

Title: Biomimetic synthesis of nanostructured $\text{WO}_3 \cdot \text{H}_2\text{O}$ particles and subsequent thermal conversion to WO_3

Manuscript ID: RSOS-182137

The editor assigned to your manuscript has now received comments from reviewers. We would

like you to revise your paper in accordance with the referee and Subject Editor suggestions which can be found below (not including confidential reports to the Editor). Please note this decision does not guarantee eventual acceptance.

Please submit your revised paper before 21-Apr-2019. Please note that the revision deadline will expire at 00.00am on this date. If we do not hear from you within this time then it will be assumed that the paper has been withdrawn. In exceptional circumstances, extensions may be possible if agreed with the Editorial Office in advance. We do not allow multiple rounds of revision so we urge you to make every effort to fully address all of the comments at this stage. If deemed necessary by the Editors, your manuscript will be sent back to one or more of the original reviewers for assessment. If the original reviewers are not available we may invite new reviewers.

On behalf of the Subject Editor Professor Anthony Stace and the Associate Editor Professor Tobias Hertel.

RSC Associate Editor:
Comments to the Author:
(There are no comments.)

RSC Subject Editor:
Comments to the Author:
(There are no comments.)

Reviewers' Comments to Author:

Reviewer: 1

Comments to the Author(s)

The present manuscript reports biomimetic synthesis of nanostructured $\text{WO}_3 \cdot \text{H}_2\text{O}$ particles. The synthesis/characterization techniques are performed at a high technical standard, and discussion given in the manuscript are basically sound. In this context, this manuscript is potentially acceptable, according to the publication criteria of Royal Society Open Science. However, I think, further discussions should be added to show clearly the back ground of the present study, and give robustness to the conclusions.

I'd like to add following remarks to be further considered by the authors;

- 1) I wonder if the present manuscript is the first report on the biomimetic synthesis of $\text{WO}_3 \cdot \text{H}_2\text{O}$ (and WO_3) or not. Otherwise, what's a significant advance from the previous biomimetic synthesis of WO_3 -based materials? The advance of the present work should be clearly discussed in the introduction section; significant advance of scientific knowledge is a criteria of acceptance by RSOS.
- 2) The authors describe their previous works on the synthesis of CeO_2 and SnO in the introduction section (the second paragraph in the introduction). The context somehow gives us an impression that the present work had been routinely-done. I suggest to add some comments to describe scientific challenges for the biomimetic synthesis of WO_3 , and how the present system is different from previous ones.
- 3) The crystallization in the present system is highly sensitive to the pH of the solution. pH after the reaction as well as before the reaction should be given to confirm robustness of the discussion; precipitation of crystals is usually accompanied by the change of pH of reaction mixture.
- 4) What is the aging time employed for samples appeared in Figs 1, 2, 3, 4, 5, 6, and 7. The authors describes in the experimental section that they tested aging time 1-7 days. Which aging time was chosen for the discussions in respective figures?
- 5) Reaction kinetics of the crystallization process should be discussed on at least one sample. I'm wondering how the crystallization occurs as a function of reaction time. Is there any induction period for the crystallization? How does the morphology develop with reaction time?
- 6) The weight ratio of gelatin/ WO_3 should be calculated from the TG data of the obtained product. This information gives us an idea on what kind of composite is indeed obtained.
- 7) Fig4: "the flat face of the layered structure" is ambiguous. It should be marked in Fig.4.
- 8) It might be interesting to compare TEM images and diffractions before and after the calcination. Is there any relevance of crystallographic orientation between them?

Reviewer: 2

Comments to the Author(s)

The authors report the new biomimetic synthesis of $\text{WO}_3 \cdot \text{H}_2\text{O}$ particles, which can further be converted to WO_3 particles by heat treatment at higher temperature. The synthesis method and discussion on the mechanisms of the particle formation are technically sound, and the conclusions are supported by experimental data obtained from sufficient number of samples. Therefore the manuscript would be publishable in Royal Society Open Science after addressing minor issues shown below.

The inner structure of the particles should be analyzed. Are the size and shape of the pores, the crystalline size and orientation the same on the surface and inside of the particles? Surface area,

pore volume, pore size distribution, true density, particle density, etc. are also importance to be measured for detailed characterization.

In the Introduction section, the authors mentioned that WO₃ has been used in various devices. Thus, readers would be interested in what sort of advantages the newly prepared WO₃ particles have or which application field they can be practically used in.

Compared with the other published works on biomimetic synthesis, the method and findings in the manuscript sound not quite new. Thus, a little more characterization or property measurement of the obtained particles would further demonstrate importance of the work.

Author's Response to Decision Letter for (RSOS-182137.R0)

See Appendix A.

Decision letter (RSOS-182137.R1)

10-May-2019

Dear Dr Uchiyama:

Title: Biomimetic synthesis of nanostructured WO₃·H₂O particles and subsequent thermal conversion to WO₃

Manuscript ID: RSOS-182137.R1

It is a pleasure to accept your manuscript in its current form for publication in Royal Society Open Science. The chemistry content of Royal Society Open Science is published in collaboration with the Royal Society of Chemistry.

On behalf of the Subject Editor Professor Anthony Stace and the Associate Editor Professor Tobias Hertel.

RSC Associate Editor
Comments to the Author:
(There are no comments.)

Reviewer(s)' Comments to Author:

Appendix A

Our response to Reviewer 1

Thank you for your careful reading and positive review.

Referee 1's original comment

The present manuscript reports biomimetic synthesis of nanostructured $\text{WO}_3 \cdot \text{H}_2\text{O}$ particles. The synthesis/characterization techniques are performed at a high technical standard, and discussion given in the manuscript are basically sound. In this context, this manuscript is potentially acceptable, according to the publication criteria of Royal Society Open Science. However, I think, further discussions should be added to show clearly the back ground of the present study, and give robustness to the conclusions.

I'd like to add following remarks to be further considered by the authors;

Referee 1's Question 1:

1) I wonder if the present manuscript is the first report on the biomimetic synthesis of $\text{WO}_3 \cdot \text{H}_2\text{O}$ (and WO_3) or not. Otherwise, what's a significant advance from the previous biomimetic synthesis of WO_3 -based materials? The advance of the present work should be clearly discussed in the introduction section; significant advance of scientific knowledge is a criteria of acceptance by RSOS.

Our response 1:

Recently, many types of biomimetic synthesis of inorganic materials have been suggested, where the factors incorporated from biomineralization vary from work to work. Many of the biomimetic works of functional metal oxide materials focus on the similarity in the “resultant nanostructure” between the products and the real biominerals, and the resultant device performance.

On the other hand, our groups have studied “new aqueous technique” mimicking biomineralization for nanostructural control of materials. We think that the key factors of

biomineralization are (1) the interaction between inorganic crystals and biological polymers, and (2) the multistep synthetic procedure via metastable phases as the precursor materials, and have suggested new approaches containing the one or both factors for making nanostructures like biominerals. The “biomimetic synthetic route” would lead to the formation of “biomimetic nanostructure” which is the hierarchical structures consisting of oriented inorganic nanocrystallites.

In the present work, the biomimetic aqueous route with biological polymers has been suggested for preparing WO_3 materials. The synthetic method with a biological polymer, gelatin, is a new and original concept for WO_3 -based materials. Moreover, the WO_3 products of the present work had the hierarchical nanostructures like biominerals (the layered structure consisting of crystallographically-oriented nanorods).

We believe that these concept and results of the present work have a fully significant advance and this manuscript is worthy to be published in Royal Society Open Science.

In order to clarify the advance of the present work, in pp. 1, line 37, the introduction section has been modified as follows;

“Biomimetic structures have been widely made from nanoscale inorganic units and biological polymers [11-13, 18-25]. Many works about biomimetic synthesis of functional metal oxide materials mainly focused on the similarity in the resultant nanostructure between the products and the real biominerals, and the resultant device performance. On the other hand, we have focused on “biomimetic synthetic route”, and attempted to construct novel aqueous techniques for making nanostructured materials. We think that the key factors of biomineralization are (1) the interaction between inorganic crystals and biological polymers, and (2) the multistep synthetic procedure via metastable phases as the precursor materials, and have suggested new approaches containing the one or both factors for making hierarchical structures consisting of oriented nanocrystallites like biominerals. Cocoon-like CeCO_3OH particles consisting of nanoscale crystallites were prepared from aqueous solutions and gels containing CeCl_3 and biological polymers such as gelatin and agar, by the addition of $(\text{NH}_4)_2\text{CO}_3$ solutions. They were then thermally converted to CeO_2 particles with the same morphologies [26]. Spherical SnO particles consisting of radially-branched platy

units were produced by aging $\text{Sn}_6\text{O}_4(\text{OH})_4$ in aqueous solutions containing gelatin at 60 °C [27]. Hence, biomimetic aqueous routes are promising ways to fabricate nanostructured inorganic materials.”

Referee 1’s Question 2:

2) The authors describe their previous works on the synthesis of CeO_2 and SnO in the introduction section (the second paragraph in the introduction). The context somehow gives us an impression that the present work had been routinely-done. I suggest to add some comments to describe scientific challenges for the biomimetic synthesis of WO_3 , and how the present system is different from previous ones.

Our response 2:

We have previously reported several works about the biomimetic synthesis of inorganic materials. As mentioned in the response 1, we think that the key factors of biomineralization are (1) the interaction between inorganic crystals and biological polymers, and (2) the multistep synthetic procedure via metastable phases as the precursor materials. The suitable route for making a biomimetic nanostructure varies from one compound to another, and thus we have to find the adequate method depending on the target materials. Thus, our papers always contain a new insight, and thus are not routine works.

The present work mainly focused on the nanostructural control of precursor materials ($\text{WO}_3 \cdot \text{H}_2\text{O}$) with biological polymers and the thermal conversion to metal oxide materials (WO_3). As mentioned in the introduction section, we have reported the biomimetic synthesis of nanostructured CeO_2 materials, which was done on the basis of a similar strategy. We achieved the preparation of nanostructured CeCO_3OH particles with biological polymers (gelatin and agar) and the thermal conversion to CeO_2 , while the crystallographic orientation of inorganic units like

biominerals were not observed in the CeO_2 products. Thus, in the present work, we selected $\text{WO}_3 \cdot \text{H}_2\text{O}$ and WO_3 for making hierarchical structures consisting of oriented inorganic nanocrystallites like biominerals. The topotactic transformation from $\text{WO}_3 \cdot \text{H}_2\text{O}$ to WO_3 crystals has been reported, which would allow us to keep the crystallographic orientation of inorganic units after the thermal conversion to metal oxide.

In order to more clearly show the concept of the present work, in pp. 1, line 54, the introduction section has been modified as follows;

“Here, nanostructured $\text{WO}_3 \cdot \text{H}_2\text{O}$ particles were prepared as WO_3 precursors from $(\text{NH}_4)_{10}\text{W}_{12}\text{O}_{41} \cdot 5\text{H}_2\text{O}$ aqueous solutions that contained gelatin; the WO_3 particles were subsequently obtained by heating the precursors. As mentioned above, we have previously tried to prepare nanostructured CeO_2 materials on the basis of a similar strategy [26]. In that work, nanostructured CeCO_3OH particles were obtained with biological polymers and then thermally converted to CeO_2 , while the crystallographic orientation of inorganic units like biominerals were not observed in the CeO_2 products [26]. On the other hand, $\text{WO}_3 \cdot \text{H}_2\text{O}$ were reported to topotactically transform to monoclinic WO_3 crystals [33], which would allow us to keep the crystallographic orientation of inorganic units after the thermal conversion to metal oxide. We varied the pH and the gelatin concentration to investigate the effects on size, shape, and crystal phase of the WO_3 precursors and WO_3 particles.”

Referee 1’s Question 3:

3) The crystallization in the present system is highly sensitive to the pH of the solution. pH after the reaction as well as before the reaction should be given to confirm robustness of the discussion; precipitation of crystals is usually accompanied by the change of pH of reaction mixture.

Our response 3:

We appreciate the referee’s helpful suggestion. The precipitation of $\text{WO}_3 \cdot \text{H}_2\text{O}$ under an acidic

condition seems to be as follows;

This reaction consumes H^+ ions, resulting in the increase in the pH value.

About some conditions, we measured the pH value of the solutions before and after the reaction.

The pH value is shown as follows;

Table. pH value of the solutions before and after the reaction

pH of solvents	[gelatin] / g L ⁻¹	pH of solutions before the reaction	pH of solutions after the reaction
0.6	0	0.63	0.65
0.6	2.0	0.60	0.59
1.0	0	1.05	1.09
1.0	2.0	1.04	1.02

The addition of gelatin didn't provide a significant pH change, and the pH value almost unchanged after the precipitation. In the present case, the $[(\text{NH}_4)_{10}\text{W}_{12}\text{O}_{41} \cdot 5\text{H}_2\text{O}]$ was very low (1.7 mM), and thus the pH change was deduced to be small during the reaction.

In pp. 3, line 6, the description on pH change has been added as follows;

“The precipitation of $\text{WO}_3 \cdot \text{H}_2\text{O}$ under an acidic condition seems to be as follows;

This reaction consumes H^+ ions, resulting in the increase in the pH value. On the other hand, in the present case, the pH value almost unchanged after the precipitation. Here, the $[(\text{NH}_4)_{10}\text{W}_{12}\text{O}_{41} \cdot 5\text{H}_2\text{O}]$ was very low (1.7 mM), and thus the pH change was deduced to be small during the reaction.”

Referee 1's Question 4 and 5:

4) What is the aging time employed for samples appeared in Figs 1, 2, 3, 4, 5, 6, and 7. The authors describes in the experimental section that they tested aging time 1-7 days. Which aging time was chosen for the discussions in respective figures?

5) Reaction kinetics of the crystallization process should be discussed on at least one sample. I'm wondering how the crystallization occurs as a function of reaction time. Is there any induction period for the crystallization? How does the morphology develop with reaction time?

Our response 4 and 5:

The aging times which employed for the samples appeared in Figures 1-7 corresponds to those shown in Table 1.

In this work, at first, we performed preliminary experiments to know the reaction time to obtain sufficient amounts of products. We changed the aging time between 1-7 days at all pH and [gelatin] conditions, and knew the sufficient time that the increase in the sample yield stopped.

About the experimental condition with $C_{ge} = 2.0 \text{ g L}^{-1}$ at pH 1.0, we investigated the effect of the reaction time on the morphology and crystal phase of the $\text{WO}_3 \cdot \text{H}_2\text{O}$ particles. No precipitation was observed for 1-3 days, while yellowish precipitates appeared after 4 days. The diffraction peaks attributed to $\text{WO}_3 \cdot \text{H}_2\text{O}$ were observed irrespective of the reaction times (4-7 days). The morphology of the 4-7 days $\text{WO}_3 \cdot \text{H}_2\text{O}$ products changed with reaction time. The samples obtained on 4 days was the mixture of layered plates and spherical particles. The spherical particles disappeared with increasing reaction times, and only layered plates were observed after 6 days.

The spherical particles found in the samples of 4-5 days were thought to be the composites of tungstate ions and gelatin. As described in the experimental section, in this work, the $(\text{NH}_4)_{10}\text{W}_{12}\text{O}_{41}$ aqueous solutions immediately became cloudy by an addition of gelatin, which might to be attributed to the formation of the composites of tungstate ions and gelatin. The cloudy suspension became transparent again by stirring at 80 °C, and then was used as the precursor solutions. In the case of the samples of 4-5 days, the precipitation of $\text{WO}_3 \cdot \text{H}_2\text{O}$ didn't complete, and

thus unreacted tungstate ions remained in the solutions. The tungstate ions might precipitate as the gelatin composite during cooling, forming the spherical particles.

Thus, we set the reaction time at the sufficient one when the increase in the sample yield stopped and the spherical particles disappeared. The sufficient aging time differed depending on the experimental conditions (pH and [gelatin]), which was described in Table 1. Thus, the aging time employed for samples in Figs 1-7 corresponds to those described in Table 1.

In order to more clearly show the meaning of “aging time”, in pp. 2, line 27, the description on the aging times and product yields has been modified as follows;

“At first, we performed preliminary experiments to know the reaction time that the increase in the sample yield stopped. Aqueous solutions of 1.7 mM $[(\text{NH}_4)_{10}\text{W}_{12}\text{O}_{41}\cdot 5\text{H}_2\text{O}]$ and 0–2.0 g L⁻¹ gelatin (C_{ge}) with HCl at pH 0.6–1.0 were aged at 60 °C for 1–7 days. Yellowish WO_3 precursors were precipitated by aging irrespective of C_{ge} and pH. The aging times that the increase in the sample yield stopped, and precursor yields are listed in Table 1. The precipitation of the WO_3 precursors was slower and the yield decreased with increasing pH, which indicated that nucleation was suppressed by the decreased acidity. Tungsten oxides precipitate as hydrous tungstic acid ($\text{H}_2\text{WO}_4\cdot n\text{H}_2\text{O}$) [35-36] and tungsten trioxide ($\text{WO}_3\cdot n\text{H}_2\text{O}$) [33-34] under strongly acidic conditions, and their solubility increases with pH [28-32] In the present case, the higher solubility under more weakly acidic conditions caused a slower nucleation rate and thus a lower yield of WO_3 precursors. Moreover, the addition of gelatin also inhibited the deposition of WO_3 precursors because its amino groups might have coordinated with tungstate ions, leading to suppressed nucleation. On the basis of these results, we employed the aging time described in Table 1 for a sample preparation.”

And, figure captions has been modified as follows;

Figure 1. XRD patterns of WO_3 precursors prepared from $(\text{NH}_4)_{10}\text{W}_{12}\text{O}_{41}$ solutions with $C_{\text{ge}}=0\text{--}2.0$ g L⁻¹ and HCl at pH 0.6–1.0 (the aging time was as shown in Table 1).

Figure 2. SEM images of WO_3 precursors prepared from $(\text{NH}_4)_{10}\text{W}_{12}\text{O}_{41}$ solutions with $C_{\text{ge}}=0$ g L⁻¹

and HCl at pH 0.6 (a,b), pH 0.8 (c,d) and pH 1.0 (e,f) (the aging time was as shown in Table 1).

Figure 3. SEM images of WO_3 precursors prepared from $(\text{NH}_4)_{10}\text{W}_{12}\text{O}_{41}$ solutions with $C_{\text{ge}}=0.2 \text{ g L}^{-1}$ (a,b) and 2.0 g L^{-1} (c,d) and HCl at pH 0.6 (the aging time was as shown in Table 1).

Figure 4. SEM images of WO_3 precursors prepared from $(\text{NH}_4)_{10}\text{W}_{12}\text{O}_{41}$ solutions with $C_{\text{ge}}=0.2 \text{ g L}^{-1}$ (a) and 1.5 g L^{-1} (b–d) and HCl at pH 1.0 (the aging time was as shown in Table 1).

Figure 7. TG/DTA curves for WO_3 precursors prepared from $(\text{NH}_4)_{10}\text{W}_{12}\text{O}_{41}$ solutions with $C_{\text{ge}}=2.0 \text{ g L}^{-1}$ and HCl at pH 1.0 (the aging time was as shown in Table 1).

Figure 9. XRD patterns of heat-treated WO_3 products obtained from WO_3 precursors prepared by aging for 7 days from $(\text{NH}_4)_{10}\text{W}_{12}\text{O}_{41}$ solutions with $C_{\text{ge}}=1.5 \text{ g L}^{-1}$ and HCl at pH 1.0.

Figure 10. FE-SEM (a–b) and FE-TEM (c–d) images of heat-treated WO_3 products from WO_3 precursors prepared by aging for 7 days from $(\text{NH}_4)_{10}\text{W}_{12}\text{O}_{41}$ solutions with $C_{\text{ge}}=1.5 \text{ g L}^{-1}$ and HCl at pH 1.0.

In pp. 5, line 24, the results and discussion about the effect of aging time on the morphology and crystal phase has been added as follows;

“We investigated the effect of the aging time on the morphology and crystal phase of WO_3 precursors. WO_3 precursors were prepared by aging for 1-7 days from $(\text{NH}_4)_{10}\text{W}_{12}\text{O}_{41}$ solutions with $C_{\text{ge}} = 2.0 \text{ g L}^{-1}$ and HCl at pH 1.0. No precipitation was observed for 1-3 days, while the precipitation appeared after 4 days. Figure 5 shows the XRD patterns of WO_3 precursors prepared by aging for 4-7 days. The diffraction peaks attributed to $\text{WO}_3 \cdot \text{H}_2\text{O}$ were observed irrespective of the aging times. Figure 6 shows the SEM images of the WO_3 precursors. The precipitate obtained on 4 days was the mixture of layered plates and spherical particles. The spherical particles disappeared with increasing aging times, and only layered plates were observed after 6 days. The spherical particles found in the precipitates of 4-5 days were thought to be the composites of tungstate ions and gelatin. As described in the experimental section, in this work, the $(\text{NH}_4)_{10}\text{W}_{12}\text{O}_{41}$ aqueous solutions immediately became cloudy by an addition of gelatin, which

might to be attributed to the formation of the composites of tungstate ions and gelatin. The cloudy suspension became transparent again by stirring at 80 °C, and then was used as the precursor solutions. In the case of the 4-5 days aging, the precipitation of $WO_3 \cdot H_2O$ didn't complete, and thus unreacted tungstate ions remained in the solutions. The tungstate ions might precipitate as the gelatin composite during cooling, forming the spherical particles."

And, the XRD patterns and SEM images WO_3 precursors prepared by aging for 4-7 days from $(NH_4)_{10}W_{12}O_{41}$ solutions with $C_{ge} = 2.0 \text{ g L}^{-1}$ and HCl at pH 1.0 has been added as Figures 5 and 6.

Figure 5. XRD patterns of WO_3 precursors prepared by aging for 4-7 days from $(NH_4)_{10}W_{12}O_{41}$ solutions with $C_{ge} = 2.0 \text{ g L}^{-1}$ and HCl at pH 1.0.

Figure 6. SEM images of WO₃ precursors prepared by aging for 4 (a) and 6 (b) days from (NH₄)₁₀W₁₂O₄₁ solutions with $C_{ge} = 2.0 \text{ g L}^{-1}$ and HCl at pH 1.0.

Referee 1's Question 6:

6) The weight ratio of gelatin/WO₃ should be calculated from the TG data of the obtained product. This information gives us an idea on what kind of composite is indeed obtained.

Our response 6:

As mentioned in pp. 7, line 21, the weight loss attributed to the combustion of residual gelatin at 550 °C was ca. 1 wt%. Moreover, we performed FT-IR analysis for the WO₃·H₂O sample, where no absorption peaks due to gelatin were not seen. Thus, only a little amount of gelatin was thought to remain on the WO₃·H₂O products.

FT-IR spectra has been added as Figure 8 into the main text as follows;

Figure 8. FT-IR spectra for WO_3 precursors prepared from $(\text{NH}_4)_{10}\text{W}_{12}\text{O}_{41}$ solutions with $C_{\text{ge}} = 0$ and 2.0 g L^{-1} and HCl at pH 1.0 (the aging time was as shown in Table 1).

And, in pp. 7, line 21, the description on the FT-IR analysis has been added as follows;

“In addition, a slight weight loss of 1 wt% was detected at $550 \text{ }^\circ\text{C}$, which could be attributed to combustion of residual gelatin. We also investigated the presence of residual gelatin on the WO_3 precursors by FT-IR analysis. Figure 8 shows FT-IR spectra for the WO_3 precursors ($C_{\text{ge}} = 0$ and 2.0 g L^{-1} , pH 1.0). No absorption peaks due to gelatin were not detected, which also indicates that only a little amount of gelatin remained on the $\text{WO}_3 \cdot \text{H}_2\text{O}$ products.”

Referee 1’s Question 7:

Fig4: “the flat face of the layered structure” is ambiguous. It should be marked in Fig.4.

Our response 7:

In order to clearly show “the flat face of the layered structure”, Figures 4 and 10 was modified as follows;

Figure 4. SEM images of WO_3 precursors prepared from $(\text{NH}_4)_{10}\text{W}_{12}\text{O}_{41}$ solutions with $C_{\text{ge}}=0.2 \text{ g L}^{-1}$ (a) and 1.5 g L^{-1} (b–d) and HCl at pH 1.0 (the aging time was as shown in Table 1).

Figure 10. FE-SEM (a–b) and FE-TEM (c–d) images of heat-treated WO_3 products from WO_3 precursors prepared by aging for 7 days from $(\text{NH}_4)_{10}\text{W}_{12}\text{O}_{41}$ solutions with $C_{\text{ge}}=1.5 \text{ g L}^{-1}$ and HCl at pH 1.0.

Referee 1's Question 8:

8) It might be interesting to compare TEM images and diffractions before and after the calcination.

Is there any relevance of crystallographic orientation between them?

Our response 8:

Thank you for a good suggestion. As the referee 1 mentioned, the comparison of TEM images between the samples before and after calcination is definitely an interesting issue.

In fact, we attempted to evaluate the crystallographic orientation of the $\text{WO}_3 \cdot \text{H}_2\text{O}$ precursor materials, and to discuss the relevance to the resultant WO_3 materials. However, the $\text{WO}_3 \cdot \text{H}_2\text{O}$ crystals were not stable toward an electron beam, and deformed during the TEM analysis. Thus, we have not yet obtained the adequate information about the crystallographic orientation.

On the other hand, as discussed in the XRD analysis part, the flat face of $\text{WO}_3 \cdot \text{H}_2\text{O}$ layered plates was found to be (010) plane of the $\text{WO}_3 \cdot \text{H}_2\text{O}$ crystal. Moreover, the electron diffraction pattern of WO_3 (Fig. 7d) shows that the flat face of heat-treated WO_3 layers is (001) plane of monoclinic WO_3 . $\text{WO}_3 \cdot \text{H}_2\text{O}$ were reported to topotactically transform to monoclinic WO_3 crystals [33]. These suggest the topotactic transformation of [010]-oriented $\text{WO}_3 \cdot \text{H}_2\text{O}$ crystals to [001]-oriented WO_3 .

In order to clarify the crystallographic relevance between the samples before and after calcination, in pp. 8, line 56, the discussion part about the TEM analysis has been modified as follows;

“Figure 10 shows FE-SEM and FE-TEM images of the heat-treated WO_3 products ($C_{ge} = 1.5 \text{ g} \cdot \text{L}^{-1}$, pH 1.0). As shown in figure 10a, the layered structure of the WO_3 precursors remained after thermal conversion to WO_3 . In addition, orthogonally-crossed nanorods 50 nm in width were observed in the WO_3 layers (figures 10b,c). Regular diffraction spots were observed in the selected area electron diffraction pattern of the WO_3 (figure 10d). The diffraction spots indicated that the rod-like units were oriented in the same crystallographic direction, and the flat face of heat-treated WO_3 layers is (001) plane of monoclinic WO_3 . $\text{WO}_3 \cdot \text{H}_2\text{O}$ were reported to topotactically transform to monoclinic WO_3 crystals [33]. As discussed in the XRD measurement of WO_3 precursors, the flat face of the $\text{WO}_3 \cdot \text{H}_2\text{O}$ layered structures was the (010) plane of $\text{WO}_3 \cdot \text{H}_2\text{O}$ crystals. These suggest the topotactic transformation of [010]-oriented $\text{WO}_3 \cdot \text{H}_2\text{O}$ layered plates to [001]-oriented WO_3 .”

Thank you for your careful reading and positive review.

Our response to Reviewer 2

Thank you for your careful reading and positive review.

Referee 2's original comment

The authors report the new biomimetic synthesis of $\text{WO}_3 \cdot \text{H}_2\text{O}$ particles, which can further be converted to WO_3 particles by heat treatment at higher temperature. The synthesis method and discussion on the mechanisms of the particle formation are technically sound, and the conclusions are supported by experimental data obtained from sufficient number of samples. Therefore the manuscript would be publishable in Royal Society Open Science after addressing minor issues shown below.

Referee 2's Question 1:

The inner structure of the particles should be analyzed. Are the size and shape of the pores, the crystalline size and orientation the same on the surface and inside of the particles? Surface area, pore volume, pore size distribution, true density, particle density, etc. are also importance to be measured for detailed characterization.

Our response 1:

Thank you for a good suggestion. As the referee 1 mentioned, the detailed analysis of the porous structures of the WO_3 products is very important.

We attempted to observe the inside structure of the products. However, the unit crystallites were densely packed in the layered architectures, and thus the inner structure was hard to be evaluated by SEM and TEM analysis. The detailed analysis is our important remaining issue.

Although the direct observation of the pore and crystallite sizes was not achieved, we evaluated the BET surface area of the WO_3 products by N_2 adsorption method. The surface area of the layered structure obtained by an addition of gelatin ($C_{\text{ge}} = 2.0 \text{ g L}^{-1}$) was $5.02 \text{ m}^2 \text{ g}^{-1}$, which was larger than

that of the random aggregates of $C_{ge} = 0 \text{ g L}^{-1}$ ($2.31 \text{ m}^2 \text{ g}^{-1}$). These shows that the synthesis route with biological polymer is effective for preparing nanostructured materials.

In pp. 9 line 5 the description on the surface area of the WO_3 products has been added as follows;

“We evaluated the BET surface area of the WO_3 products by N_2 adsorption method. The surface area of the layered structure obtained by an addition of gelatin ($C_{ge} = 1.5 \text{ g-L}^{-1}$, pH 1.0) was $5.02 \text{ m}^2 \text{ g}^{-1}$, which was larger than that of the random aggregates of $C_{ge} = 0 \text{ g-L}^{-1}$, pH 1.0 ($2.31 \text{ m}^2 \text{ g}^{-1}$).”

Referee 2’s Question 2:

In the Introduction section, the authors mentioned that WO_3 has been used in various devices. Thus, readers would be interested in what sort of advantages the newly prepared WO_3 particles have or which application field they can be practically used in.

Our response 2:

As mentioned in the response 1, the addition of gelatin resulted in the increase in the surface area of WO_3 particle materials. Such WO_3 materials with a larger surface area are thought to be suitable for the photoelectrodes and gas sensors. Since photoelectrochemical reactions and gas sensing occur on the surface of the electrode materials, a larger surface area of the nanostructured electrodes would result in better device performance.

In order to clarify the advantage of the WO_3 products, in pp. 9, line 3, the discussion part has been modified as follows;

“These results suggested that the WO_3 layered structures prepared with gelatin have highly ordered nanostructures consisting of oriented inorganic nanoscale units like biominerals such as nacles, sea urchin spines, and eggshells [9-17]. We evaluated the BET surface area of the WO_3 products by N_2 adsorption method. The surface area of the layered structure obtained by an addition of gelatin ($C_{ge} = 1.5 \text{ g-L}^{-1}$, pH 1.0) was $5.02 \text{ m}^2 \text{ g}^{-1}$, which was larger than that of the random aggregates of $C_{ge} = 0 \text{ g-L}^{-1}$, pH 1.0 ($2.31 \text{ m}^2 \text{ g}^{-1}$). The hierarchical nanostructures are thought to be suitable for photoelectrode and gas sensor materials.”

Now, we are trying to prepare WO_3 film materials on the basis of the present work, and to evaluate the device performance. We would report the results in the near future.

Referee 2's Question 3:

Compared with the other published works on biomimetic synthesis, the method and findings in the manuscript sound not quite new. Thus, a little more characterization or property measurement of the obtained particles would further demonstrate importance of the work.

Our response 3:

As the referee 2 mentioned, recently, many types of biomimetic synthesis of inorganic materials have been suggested. Many of them focus on the similarity in the “resultant nanostructure” between the products and the real biominerals. On the other hand, our groups have studied new “synthetic route” mimicking biomineralization. We think that the key factors of biomineralization are (1) the interaction between inorganic crystals and biological polymers, and (2) the multistep synthetic procedure via metastable phases as the precursor materials, and have suggested new approaches containing the one or both factors for making nanostructures like biominerals. The “biomimetic synthetic route” would lead to the formation of “biomimetic nanostructure” which is the hierarchical structures consisting of oriented inorganic nanocrystallites.

In the present work, the biomimetic aqueous route with biological polymers has been suggested for preparing WO_3 materials. The synthetic method with a biological polymer, gelatin, is a new and original concept for WO_3 -based materials. Moreover, the WO_3 products of the present work had the hierarchical nanostructures like biominerals (the layered structure consisting of crystallographically-oriented nanorods).

We believe that these concept and results of the present work have a fully significant advance and this manuscript is worthy to be published in Royal Society Open Science.

In order to clarify the advance of the present work, in pp. 1, line 37, the introduction section has been modified as follows;

“Biomimetic structures have been widely made from nanoscale inorganic units and biological

polymers [11-13, 18-25]. Many works about biomimetic synthesis of functional metal oxide materials mainly focused on the similarity in the resultant nanostructure between the products and the real biominerals, and the resultant device performance. On the other hand, we have focused on “biomimetic synthetic route”, and attempted to construct novel aqueous techniques for making nanostructured materials. We think that the key factors of biomineralization are (1) the interaction between inorganic crystals and biological polymers, and (2) the multistep synthetic procedure via metastable phases as the precursor materials, and have suggested new approaches containing the one or both factors for making hierarchical structures consisting of oriented nanocrystallites like biominerals. Cocoon-like CeCO_3OH particles consisting of nanoscale crystallites were prepared from aqueous solutions and gels containing CeCl_3 and biological polymers such as gelatin and agar, by the addition of $(\text{NH}_4)_2\text{CO}_3$ solutions. They were then thermally converted to CeO_2 particles with the same morphologies [26]. Spherical SnO particles consisting of radially-branched platy units were produced by aging $\text{Sn}_6\text{O}_4(\text{OH})_4$ in aqueous solutions containing gelatin at 60 °C [27]. Hence, biomimetic aqueous routes are promising ways to fabricate nanostructured inorganic materials.”

Thank you for your careful reading.